# The Prevalence and Molecular Landscape of Lynch Syndrome in the Affected and General Population

**DOI:** 10.3390/cancers15143663

**Published:** 2023-07-18

**Authors:** Laura Roht, Piret Laidre, Mikk Tooming, Neeme Tõnisson, Margit Nõukas, Miriam Nurm, Hanno Roomere, Kadri Rekker, Kadri Toome, Olga Fjodorova, Ülle Murumets, Ustina Šamarina, Sander Pajusalu, Anu Aaspõllu, Liis Salumäe, Kristina Muhu, Jaan Soplepmann, Katrin Õunap, Tiina Kahre

**Affiliations:** 1Department of Clinical Genetics, Institute of Clinical Medicine, University of Tartu, 50406 Tartu, Estoniasander.pajusalu@kliinikum.ee (S.P.);; 2Department of Clinical Genetics, Genetics and Personalized Medicine Clinic, Tartu University Hospital, 50406 Tartu, Estonia; 3Department of Laboratory Genetics, Genetics and Personalized Medicine Clinic, Tartu University Hospital, 50406 Tartu, Estonia; 4Estonian Biobank, Institute of Genomics, University of Tartu, 51010 Tartu, Estonia; 5Institute of Molecular and Cell Biology, University of Tartu, 51010 Tartu, Estonia; 6Asper Biogene LLC, 50410 Tartu, Estonia; 7Pathology Service, Tartu University Hospital, 50406 Tartu, Estonia; 8Estonian Unemployment Insurance Fund, 10142 Tallinn, Estonia; 9Department of Surgical and Gynecological Oncology, Surgery Clinic, Tartu University Hospital, 50406 Tartu, Estonia; 10Department of Hematology and Oncology, Institute of Clinical Medicine, University of Tartu, 50406 Tartu, Estonia

**Keywords:** Lynch syndrome, cancer genetics, mismatch repair (MMR) genes, immunohistochemistry (IHC)

## Abstract

**Simple Summary:**

Lynch syndrome accounts for 2–3% of all CRC cases. This retroactive and prospective study aimed to estimate the prevalence of Lynch syndrome and describe disease-causing variants in mismatch repair genes in a diagnostic setting and in the Estonian general population. For that, ten years (2012–2022) of data was gathered. In addition, a pilot study for estimating the improvement of Lynch syndrome diagnostics by raising the age limit of mismatch repair gene immunohistochemistry was conducted. We estimated the birth prevalence of LS in Estonia at 1:8638 (95% CI: 1:9859–7588) or 11.58 (95% CI: 10.14–13.18) for 100,000 LBs between 1930 and 2003. The prevalence of Lynch syndrome has increased approximately six-fold in ten years. Due to the improvement of awareness in families and patients sharing information with their family members, the latter receives the diagnosis eight years earlier when most individuals are still healthy, illustrating the benefit of genetic testing and therefore an opportunity for prevention. Furthermore, the pilot study proved to be beneficial.

**Abstract:**

Background: Lynch syndrome (LS) is the most frequent genetically pre-disposed colorectal cancer (CRC) syndrome, accounting for 2–3% of all CRC cases. In Estonia, ~1000 new cases are diagnosed each year. This retroactive and prospective study aimed to estimate the prevalence of LS and describe disease-causing variants in mismatch repair (MMR) genes in a diagnostic setting and in the Estonian general population. Methods: LS data for the diagnostic cohort were gathered from 2012 to 2022 and data for the general population were acquired from the Estonian Biobank (EstBB). Furthermore, we conducted a pilot study to estimate the improvement of LS diagnostic yield by raising the age limit to >50 years for immunohistochemistry analysis of MMR genes. Results: We estimated LS live birth prevalence between 1930 and 2003 in Estonia at 1:8638 (95% CI: 1: 9859–7588). During the study period, we gathered 181 LS individuals. We saw almost a six-fold increase in case prevalence, probably deriving from better health awareness, improved diagnostic possibilities and the implementation of MMR IHC testing in a broader age group. Conclusion: The most common genes affected in the diagnostic and EstBB cohorts were *MLH1* and *PMS2* genes, respectively. The LS diagnosis mean age was 44.8 years for index cases and 36.8 years (*p* = 0.003) for family members. In the MMR IHC pilot study, 29% had LS.

## 1. Introduction

According to World Cancer Research Fund International, colorectal cancer (CRC) is among the most common cancers in the world and the second leading cause of cancer-related death. It is the third most common cancer for men and second for women (https://www.wcrf.org/, accessed on 4 July 2023). In Estonia, according to the Estonian National Institute for Health Development, among cancers, CRC is in third place for both women and men ((https://tai.ee/en, accessed on 13 March 2023). In 2020, there were more than 1.9 million new CRC cases worldwide and 935,173 people died because of it (https://www.wcrf.org/, accessed on 4 July 2023).

Genetic predisposition (monogenic or familial CRC) plays a part in up to 20% of CRCs [1]. Lynch syndrome (LS), which historically is also known as hereditary non-polyposis colorectal cancer (HNPCC), is the most frequent hereditary colorectal cancer syndrome. According to the literature, LS constitutes 2–3% of all CRCs [2]. LS is inherited in an autosomal-dominant way and caused by germline disease-causing variants in one of the four mismatch repair (MMR) genes (*MLH1, MSH2, MSH6, PMS2*) [3] or the epithelial cell adhesion molecule (*EPCAM*) gene. According to the literature, the population prevalence of pathogenic variants in MMR genes has been estimated to be 0.051% (1:1946) for *MLH1*, 0.035% (1:2841) for *MSH2*, 0.132% (1:758) for *MSH6* and 0.140% (1:714) for *PMS2* [4]. Cancer risks and, therefore, surveillance are dependent on many factors, such as MMR gene affected, sex, family history, etc. [5]. Worldwide, thousands of disease-predisposing variants in MMR genes have been reported (Clinvar database, https://www.ncbi.nlm.nih.gov/clinvar/, accessed on 13 March 2023). In 2015, Ponti and coworkers showed that, in some populations, up to 50% of MMR disease-causing variants can be founder mutations [6]. At the moment, we do not have that information on Estonia. Moreover, from molecular point of view, in cases of suspected Lynch syndrome but with no disease-causing variants found in MMR or *EPCAM* genes and with a loss of MLH1 expression in tumor tissue, up to 10% of the cases are caused by constitutional *MLH1* epimutations [7].

The DNA MMR system helps to ensure genomic integrity [8]. The main task of this system is to correct base–base mismatches and small insertion–deletion loops (indels) mostly generated during DNA replication [9].

According to the Estonian National Institute for Health Development data, in 2020, there were 965 new CRC cases (men and women) in Estonia, and it is increasing yearly (https://tai.ee/en, accessed on 13 March 2023). Therefore, approximately 28–38 individuals may have LS instead of sporadic CRC. We have difficulty identifying the majority of LS patients due to the absence of such registries, and we need more knowledge on the prevalence of LS in Estonia.

Previously, we have estimated the prevalence of LS-associated MMR gene variants to be 1:485 (Poisson 95% CI: 1:263–1:1009) in the general population based on Estonian Biobank (EstBB) data. Therefore, the primary purpose of this study was to collect information on LS prevalence in routine diagnostic settings and the general population in Estonia. Secondly, we aimed to compare the distribution of MMR gene pathogenic variants in the Estonian population with data in the literature [10]. Thirdly, we hypothesized that raising the age limit of MMR immunohistochemistry (IHC) testing in colorectal cancer patients from 50 to 70 years would improve the diagnosis of LS by increasing the diagnostic yield.

## 2. Materials and Methods

### 2.1. Study Group

We collected data on LS patients and pre-symptomatic carriers for ten years, from 2012–2022, both retroactively and prospectively, using databases of different diagnostic laboratories and referral visits at the Genetics and Personalized Medicine Clinic (GPM Clinic) of Tartu University Hospital (TUH) in Estonia. We first reviewed the molecular database of the GPM Clinic (more than 5000 samples analyzed for germline hereditary cancer risk) and our collaborators’ databases (Asper Biogene, EstBB), then collected clinical data and formed two cohorts: diagnostic and the EstBB cohort representing the Estonian general population.

### 2.2. Diagnostic Cohort

The diagnostic cohort consisted of 119 individuals. Index cases (*n* = 71) were either patients with cancer in case history or healthy carriers who were first consulted by a clinical geneticist in a Lynch family. All index cases were diagnosed in the routine clinical setting in the GPM Clinic. The remaining (*n* = 48) were family members. Details are covered in Table 1.

### 2.3. EstBB Cohort

The EstBB is a population-based biobank with over 200,000 participants’ genetic and health data. The EstBB represents more than 20% of Estonia’s adult population. All participants at the EstBB have been genotyped using the Illumina Infinum Global Screening Array (GSA, Illumina Inc., San Diego, CA, USA). A subset of participants has high-coverage sequencing data (genome and exome sequencing). All EstBB participants have provided broad written consent, allowing EstBB to continuously update participants’ health data through periodical linking to national electronic databases and recontact participants.

In the EstBB cohort, LS-associated pathogenic and likely pathogenic (P/LP) variants were identified from high-coverage genome sequencing (*n* = 2420) and exome sequencing (*n* = 2356) data. High-coverage sequencing data includes 4776 individuals, of whom 47% were female and 53% male. In addition, P/LP variants of MMR genes were searched among 201,134 EstBB participants with imputed genotyping array data. A total of 63% (*n* = 131,769) of EstBB participants were female.

Initially, we detected LS-associated disease-causing gene variants in 69 individuals from the EstBB cohort, further divided into two groups. The first group consisted of 30 individuals, 17 of whom were included in our study, who were consulted by a clinical geneticist and whose molecular findings were validated in the GPM Clinic. During consultations, we discovered that three of those 17 cases were family members. The remaining cases (13/30) were excluded from this study because the clinical geneticist had already consulted them (therefore, they were already part of the diagnostic cohort), they were unreachable by mail or phone or they refused to participate. The second group of 39 individuals from the EstBB cohort carrying LS-associated disease-causing gene variants were excluded from this study at that point mainly due to the fact that they had not yet been consulted by a clinical geneticist, and thus, molecular findings had not been validated in the GPM Clinic’s laboratory. Recalling these individuals is planned in the near future. Together with the second group, additional eight individuals, who are not already a part of the diagnostic cohort, are recalled. Details of this cohort are shown in Appendix A (Appendix A), where all disease-causing variants found in the EstBB cohort can also be seen. It has been shown in previous EstBB studies that approximately 60% of Biobank participants are interested in further investigations and will come to genetic counseling [11].

Finally, for reference and comparing clinical data, we used the detailed information of 23 individuals (fourteen index cases and an additional nine family members) from the EstBB cohort, who were all consulted by a clinical geneticist, similarly to individuals consulted in the clinical setting in the GPM Clinic, and molecular findings were validated in our medical laboratory from new blood samples.

### 2.4. MMR IHC Pilot Study

We also conducted a pilot study on the IHC testing of MMR genes in the Pathology Service of TUH, which started in January 2018, lasted until December 2021 (included) and covered CRC patients in South Estonia aged up to 70 years. Altogether, 464 MMR IHC analyses (one MMR IHC test contained four stainings) were performed during four years (100–128 per year). As a result, 19 individuals were called out for genetic consultation due to changes in MMR IHC and tested for LS.

## 3. Methods

### 3.1. Molecular Methods

Until 2016, LS molecular testing in Estonia was performed in different laboratories using Sanger and/or Next Generation Sequencing (NGS) and Multiplex Ligation-dependent Probe Amplification (MLPA) diagnostics of MMR genes. From 2016 onward, most of the diagnostic cohort cases were investigated at the GPM Clinic by the NGS TruSight Cancer panel (TSC, 94 genes, Illumina Inc., San Diego, CA, USA), and since July 2020, with TruSight Hereditary Cancer panel (TSHC, 113 genes, Illumina Inc., San Diego, CA, USA). Sanger sequencing was used for family member testing to detect already known disease-causing variants in the family. Illumina MiniSeq or NextSeq platforms were used for sequencing TSC and TSHC panels. The BWA-MEM alignment algorithm was used to align raw sequencing reads to the hg19 reference genome. Genome Analysis Toolkit (GATK) was used for variant calling [12]. ACMG criteria were used to classify pathogenicity of gene variants [13]. The clinical significance of all genetic variants was assessed using ClinVar [14], InSight (InSight database), VarSome Premium (Saphetor SA, Lausanne, Switzerland) and HGMD Pro databases (HGMD Professional database, Oiagen), as well as the Genome Aggregation Database, gnomAD [15,16,17]. To detect copy number variations (CNVs) in MMR genes, MLPA [18] diagnostics for *MLH1*, *MSH2*, *MSH6*, *PMS2* and *EPCAM* genes (MRC-Holland) were routinely used when NGS panel did not reveal any P/LP LS variants until 2021. From 2021 onward, DeCon program was used to detect possible CNVs for MMR+*EPCAM* genes from NGS data [19]. Our previous work showed that around 400 MLPA analyses were performed during the five-year study period, and no new disease-causing variants were found. NGS first discovered all exon deletions and then confirmed them by MLPA analysis [20]. Due to this knowledge, MLPA is only used to confirm pathogenic CNV findings in MMR genes in our current practice. Exceptionally, due to the presence of pseudogene (*PMS2CL*), MLPA testing is still routinely used for detecting CNVs in the *PMS2* gene.

For the EstBB cohort, the library preparation methods, quality control and array genotyping have been described elsewhere [21]. Genotype calling for sequencing, array data and quality control has been described previously [11].

To annotate variants in LS-associated genes from sequencing, array genotyping and imputed data, in-house pipeline was used. LS-associated gene variants were cross-referenced with the ClinVar database, and only likely pathogenic and pathogenic variants were retrieved. Before recontacting individuals, all selected candidate variants were confirmed by Sanger sequencing from DNA samples stored in the EstBB. At first, we identified 38 probands, who carry a total of five distinct heterozygous variants in LS-associated genes (*MLH1*, *MSH2*, *MSH6*, *PMS2*).

### 3.2. Statistical Methods

The prevalence of LS for each year was defined as a ratio of the total number of patients with this syndrome diagnosed from 2012 to 2022 and the number of people living in Estonia in the same year. The birth prevalence of LS was defined as a ratio of the total number of patients with this syndrome born within the period from 1930 to 2003 and the number of live births (LBs) in Estonia during the same period. During the years 1930–2003, there were 1,433,916 LBs based on the database of Statistics Estonia (www.stat.ee, accessed on 5 July 2023). Based on the Poisson distribution, the 95% confidence interval for birth prevalence was determined [22].

A general linear model (GLM) analysis was used to estimate the prevalence and birth prevalence of LS. For the diagnosed cases, a Poisson distribution was assumed, and the logarithmic link function was applied. The only variables in the prevalence model and in the LB prevalence model were the observation year and the birth year, respectively. All the estimated calculations including the mean expected prevalence or LB prevalence rate for a specific year and the related 95% confidence intervals were computed using R version 4.2.0 [23]. A *p*-value level of less than 0.05 was used to determine statistical significance. To estimate the difference in the mean age of LS diagnosis and for first cancer in the diagnostic index and family member cases groups, the Wilcoxon rank sum test was applied.

### 3.3. MMR IHC

Immunohistochemistry of MMR genes is a widely used method to identify MMR status, which can either be IHC positive (protein expression is normal and, therefore, nuclear staining in tumor cells is detected) or negative (loss of protein expression and no staining is detected in tumor cells) [24]. Different diagnostic workflow differences exist in different diagnostic laboratories in Estonia to investigate the malfunctioning of MMR genes. For example, TUH Department of Pathology uses MMR IHC, and one test encompasses four stainings (MutL Protein Homolog 1 (MLH1), Clone ES05, Agilent Technologies Inc; Ventana anti-MSH2 (G219-1129) Mouse Monoclonal Primary Antibody, Roche; Ventana anti-MSH6 (SP93) Rabbit Monoclonal Primary Antibody, Roche; Ventana anti-PMS2 (A16-4) Mouse Monoclonal Primary Antibody, Roche).

Our pilot study was based on TUH patients, so MMR IHC technology was used for LS screening.

## 4. Results

We collected data from 181 LS carriers in Estonia during ten years, of whom 105 were females and 76 males, aged 2–92, born between 1930 and 2020. We have detailed clinical information on 139 individuals. Of these 139 individuals, 116 were part of the diagnostic cohort, and the EstBB cohort consisted of 23 individuals. An additional 39 individuals from the EstBB cohort were put on hold primarily because they were not consulted by a clinical geneticist at the time of writing this article; recalling these individuals is planned in the near future. Details of EstBB cohorts are covered in the Appendix A.

### 4.1. Results of the Detailed Clinical Information

From the group with detailed information (139 individuals), there were 82 index cases (studied first in the family but not all had cancer) and 57 family members. This cohort included 79 females and 60 males, aged 2–92, born between 1930 and 2020. In the diagnostic cohort, the mean age of LS diagnosis was 41.5 years, and the mean age of first cancer diagnosis was 44.4 years. Most individuals (20.9% (29/139)) with cancer history had one tumor, 10.8% (15/139) had two and 4.3% (6/139) had three or more in their health history. For three individuals, we lacked data and, therefore, they were excluded from the cohort with detailed clinical data. As predicted, the most frequent tumor was colorectal cancer. From an oncological point of view, 64.0% (89/139) were healthy or had benign changes (polyps in the colon or, in one case, osteomas in the mandibula). For the EstBB cohort, the mean age of LS diagnosis was 46.3 years, and all of them are healthy from oncological perspective.

### 4.2. LS prevalence in Estonia

In our calculations, we counted only adult individuals, as LS is not clinically significant in childhood. Therefore, LS birth prevalence between 1930 and 2003 in Estonia was estimated at 1:8638 (95% CI: 1:9859–7588) or 11.58 (95% CI: 10.14–13.18) for 100,000 LBs. Figure 1 illustrates the prevalence of 100,000 LBs for both cohorts. The maximum of 14.35 cases falls to the year 1972, which is expected as LS usually manifests in middle age. The prevalence of LS (cases per 100,000 persons) has risen from 0.34 (95% CI 0.22–0.52) to 1.99 (95% CI 1.54–2.57) in ten years (2012–2022) (*p* < 0.0001) (Figure 2).

### 4.3. Clinical Aspects

LS has a broad spectrum of cancers, and the risk estimates depend on the gene affected. According to the National Comprehensive Cancer Network (NCCN) guidelines, the highest cancer risks for females are for colorectal and endometrial cancer, 8.7–61.0% and 13.0–57.0%, respectively. For males, CRC is similarly the highest cancer risk, and other cancer risks are not significantly high (https://www.nccn.org/professionals/physician_gls/pdf/genetics_colon.pdf, accessed on 5 May 2023).

The diagnostic cohort consisted of 71 index cases and 48 family members. We had 35 male and 36 female index cases born between 1943 and 2020. Altogether, 50 out of 71 index cases had at least one cancer in their health history, 18 were healthy carriers and seven of them were diagnosed because of reporting incidental findings (children and young people investigated due to other health problems). In all families with incidental findings, informed consent was signed, and pre-test counseling about incidental findings and their consequences was carried out. The mean age of the first cancer was 44.8 years, and the most frequent cancer types were colorectal, prostate, breast, endometrial and bladder cancer. The mean age of LS diagnosis was 44.6 years. Almost half (52%) of the index cases, who had cancer, had been screened for LS by MMR IHC. Altogether, in 38.5% of these cases, MLH1 and PMS2 expression were negative, which can either refer to sporadic cancer of different causes or disease-causing germline variants in the *MLH1* or *PMS2* gene. Of the 48 family members, there were 27 females and 21 males. Forty (83.3%) were healthy from an oncological point of view, and eight (16.7%) had cancer in case history: in seven cases, only one cancer and, in one case, two cancers. Six of those eight individuals had had colorectal cancer at some point in their life. The mean age of the first cancer diagnosis was 43.5, which is quite similar to that of the index cases, and it was also statistically insignificant (*p* = 0.5422). All individuals in the family member group had a family history of cancer. The mean age of LS diagnosis was 36.8 years, which is 8 years earlier than for the index cases group; it was also statistically significant (*p* = 0.0035). Details of both index and family members’ cancer cases are covered in Table 1, Figure 3 and Figure 4.

The EstBB cohort consisted of 14 index cases and nine family members. Of 14 index cases, 12 were females, and two were males. All 23 individuals were healthy from an oncological point of view. The mean age of LS diagnosis for the whole EstBB cohort was 46.45 years.

### 4.4. MMR Genes Pathogenic Variants in the Estonian Population

In the diagnostic cohort’s index cases, 28.2% (20/71) carried *MLH1* disease-causing variant; 28.2% (20/71) either had *MSH2* disease-causing variants only, or together with *EPCAM* deletion in five cases; *MSH6* variants made up 19.7% (14/71) and *PMS2* pathogenic variants 23.9% (17/71) of all the cases. *MLH1* gene variant NM_000249.4: c.1976G>C p.(Arg659Pro) was the most frequent in the diagnostic cohort. In this study, we identified ten new variants in MMR genes: four in *MSH2*, two of which were together with *EPCAM* deletion; three in *MSH6*; two in *MLH1* and one in *PMS2* gene (Table 2). All of the new variants have been added to the ClinVar database. In the EstBB cohort, *PMS2* disease-causing variants were the most common, making up more than half (57.2%; 8/14); *MSH6* variants constituted 35.7% (5/14) and there was only one case of a *MSH2* disease-causing variant (7.1%). *PMS2* variant NM_000535.7: c.861_864del, p.(Arg287Serfs*19) was the most common variant found.

In the routine clinical setting NGS study published in 2022, we found four novel MMR pathogenic gene variants in CRC patients, two in *MLH1* and two in the *MSH2* gene. These have been entered into ClinVar, and details are covered in our previous publication on Estonian CRC patients’ molecular genetic landscape [20]. One patient with a novel *MSH2* variant has had more than ten cancers in 30 years and the case is described in detail in this article. Five out of six patients with three or more cancers had either *MLH1* or *MSH2* and/or *EPCAM* disease-causing variants. The sixth case, the only known constitutional mismatch repair deficiency (CMMR-D) patient in Estonia, had compound-heterozygous variants in *MSH6* gene. She is currently 25 years old. At the age of 10, she had T-lymphoblast leukemia. At the age of 17 years, she was diagnosed with severe colorectal polyposis, and laparoscopic total procto-colectomy with end ileostomy was performed. Microscopically, the polyps were mostly tubular adenomas exhibiting high-grade dysplasia. In the biggest polyp (50 mm) from the sigmoid colon, a focus of intraepithelial adenocarcinoma (pTis) was found. At 19 years, she had lymphoblastic leukemia, and at 21 years, diffuse astrocytoma of the frontal lobe [25].

### 4.5. Case Report: Highest Number of Cancers in Health History

The highest number of cancers per person in the diagnostic cohort was more than ten. It was a female proband from southern Estonia. She was 77 years at the time of LS diagnosis. She had had 14 cancers altogether in four organ systems: three CRCs, one small intestine cancer, three urothelial system cancers, three basaliomas and four of two types of other skin cancer. The first cancer was CRC at the age of 44 years. From the family history, it is known that her mother had died of endometrial cancer at the age of 41 years, two of her mother’s sisters had gastric cancers in their 60s, and the mother’s third sister had CRC at the age of 72 years. Her son, who was a carrier, had a malignant gastrointestinal stromal tumor (GIST) in the small intestine at the age of 40 years. Her two daughters were healthy non-carriers. The pedigree is depicted in Figure 5. MMR IHC from colorectal adenocarcinoma showed a loss of MSH2 and MSH6 expression, and the NGS TruSight Cancer panel discovered a disease-causing (Class 5) variant in *MSH2* NM_000251.2: c.1283_1284del p.(His428Profs*14), which had not been described in the databases at the time the analysis was performed (in 2018). As she also had different skin tumors (basaliomas, keratoacanthomas, etc.), the final diagnosis was Muir–Torre syndrome, a subtype of Lynch syndrome.

### 4.6. MMR IHC Pilot Study

The MMR IHC pilot study resulted in 52 IHC-negative individuals (11.2%) filtered out from 464 analyses. MLH1 and PMS2 negative expression were the most common findings (36.5%). Out of 52 individuals, 19 (36.5%) were called out for germline testing, 12 (23.1%) had been consulted earlier by a clinical geneticist within routine clinical work and 21 (40.4%) were not consulted due to different reasons (not reached, did not want to attend, tested for somatic variants only, etc.) (Figure 6). Eleven individuals out of 31 had variants in MMR genes; in nine cases, P/LP variant, and in two cases, variants of unknown significance (VUS). Six individuals out of 11 (54.5%) were in this group of colorectal cancer from 50 to 70 years, where raising the age limit of MMR IHC testing from 50 to 70 was expected to improve the diagnosis of LS. 

Nine individuals out of 31 (29%) with P/LP variants were diagnosed with definite LS. The two with VUS are still probable LS cases. *MSH2* gene disease-causing variants alone or with *EPCAM* deletion were the most frequent, constituting two-thirds of the findings.

## 5. Discussion

The maximum LS birth prevalence for 100,000 LBs for both cohorts was 14.35 for individuals born in 1972, as LS usually manifests in one’s 40s or 50s. The CI widened from 2015 onward, probably because of a few children who were diagnosed due to reporting incidental findings. Otherwise, they would have been diagnosed later in life. We understand that our estimated LS birth prevalence probably is biased due to the following circumstances: inevitably, we have missed some LS individuals; some have died before diagnosis and the fact that we do not test children routinely, as LS mostly becomes clinically significant only in adulthood. LS prevalence (cases per 100,000 persons) has risen almost six times in ten years, probably due to better diagnostic opportunities and health awareness among doctors. In 2020, we estimated the prevalence of LS-associated MMR gene variants to be 1:485 in the general population. Our estimated prevalence of disease-causing variants in MMR genes in the general population is lower than that reported in the world population. It has been estimated to be 1:100–1:180 in a 2020 study [26] and at 1:279 in an earlier study [27]. There has yet to be any recent data in the literature for European populations. The latest work of Zhang et al. [28] estimated the prevalence of MMR disease-causing variants in the Chinese general population to be 0.18%, similar to our estimation from 2020.

According to the literature, the distribution of MMR genes is as follows: *MLH1* variants make up 15–40%, *MSH2* 20–40%, *MSH6* 12–35%, *PMS2* 5–25% and *EPCAM* variants <10% (https://www.ncbi.nlm.nih.gov/books/NBK1211/, accessed on 5 May 2023). Earlier, the contribution of MMR genes was estimated to be ~50% for *MLH1*, ~40% for *MSH2*, 7–20% for *MSH6* and <5% for *PMS2* [29]. According to earlier estimations, our population is different regarding *PMS2* proportion. However, recent data shows that our MMR gene distribution is similar to other populations.

In the diagnostic cohort, disease-causing variants in *MLH1* and *MSH2* genes (alone or together with *EPCAM* deletion) were the most common (28.2% of index cases for both), and the most frequently detected variant was c.1976G>C in *MLH1*. In the EstBB cohort, *PMS2* disease-causing variants comprised more than half (57.1%) of all the MMR gene variants found in that cohort. In this study, we discovered ten new variants in MMR genes: four in *MSH2*, three in *MSH6*, two in *MLH1* and one in *PMS2* gene. Most of the variants frequently identified in the diagnostic cohort were also the most common in the EstBB cohorts (Table 2; Appendix A in Appendix A). All of our findings in the diagnostic cohort were first discovered on the NGS panel. No exon deletions and duplications were found with MLPA analysis, so we have stopped using this method routinely for LS diagnostics (except for the *PMS2* gene because of its pseudogene).

We had six cases with more than three cancers in the health history: five of these individuals were either carriers of *MLH1* or *MSH2* disease-causing variants, and one was a CMMR-D case with compound-heterozygous variants in the *MSH6* gene. It confirms that *MLH1* and *MSH2* give the highest cancer risks, mainly but not only limited to CRC incidence, and as presented in the clinical case earlier in this article, *MSH2* is known to be more connected to extra-intestinal presentations than other LS genes [30,31].

Regarding the mean age of LS diagnosis, family members are diagnosed, on average, eight years earlier than index cases. This is statistically significant and probably the result of communication and awareness of potential cancer risks in the family, which brings them to the medical system. We did not find a statistically significant difference in the mean age of the first cancer diagnosis between the diagnostic cohort’s index cases and family members, which was an expected outcome. In the diagnostic cohort, on average, 2.1 family members were tested per index case. The EstBB cohort had 1.07 family members for one index case.

As Hoogerbrugge et al. presented in 2017 in their thesis of European Society of Human Genetics Conference, raising the age limit for MMR IHC testing improves the efficacy of LS diagnosing (Retrieved 4 April 2023 from www.sciencedaily.com/releases/2017/05/170529085619.htm). In our study, most LS cases were diagnosed between 50 and 70 years of age. The question is no longer whether we should screen all CRC cancers by MMR IHC or other equivalent methods, but rather who should be tested for germline disease-causing variants after the screening. In Estonia, the working group of LS, including geneticists, surgeons, oncologists, gynecologists and pathologists, has recently decided that MMR IHC will be conducted for all CRC patients irrespective of their age, and depending on the result, studied further either by a geneticist or other specialists when LS is suspected.

Our study limitations: One of our main limitations is the retrospective design of the study, due to which we have probably not been able to detect all of the LS carriers in Estonia. Before 2015, LS analyses were performed in three different genetic laboratories or sent abroad to commercial laboratories for testing. Secondly, we did not manage to involve one of the commercial genetic testing laboratories. Thirdly, some patients probably have died before testing was available or clinicians thought about this hypothesis. In addition, we do not have a national or hospital-based registry in Estonia nor a different diagnosis code in the health system, which is problematic in many countries.

## 6. Conclusions

The birth prevalence of LS in Estonia was estimated at 1:8638 (95% CI: 1:9859–7588) or 11.58 (95% CI: 10.14–13.18) for 100,000 LBs between 1930 and 2003. The prevalence of LS has increased approximately six times in ten years, illustrating the improvement in the health awareness of LS among medical doctors and better diagnostic opportunities, including an increased age range in the MMR IHC pilot study.

Surveillance and/or prophylactic surgery is the main key and end goal for preventing cancer and death from any LS spectrum cancer. Due to the improvement of LS awareness in families and patients sharing information with their family members, the latter receives the diagnosis eight years earlier, when most individuals are still healthy, illustrating the benefit of communication and awareness for earlier prevention.

## Figures and Tables

**Figure 1 cancers-15-03663-f001:**
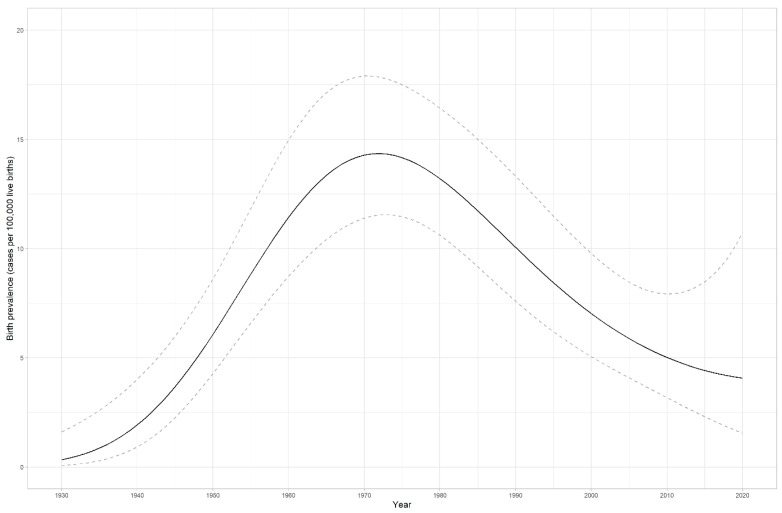
Lynch syndrome’s birth prevalence (cases per 100,000 live births) in the diagnostic and EstBB cohorts during the years 1930–2020 in Estonia. Confidence interval (95% CI) is shown in accented lines.

**Figure 2 cancers-15-03663-f002:**
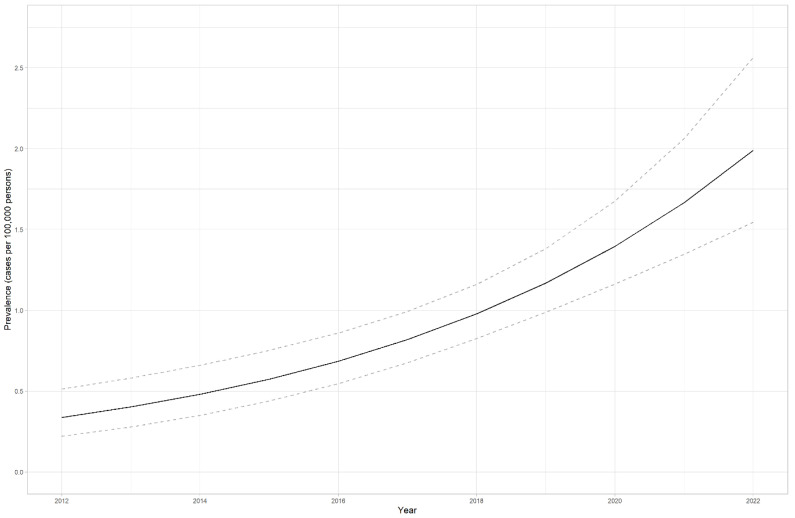
Lynch syndrome’s prevalence (cases per 100,000 persons) in Estonia from 2012 to 2022. Confidence interval (95% CI) is shown in accented lines.

**Figure 3 cancers-15-03663-f003:**
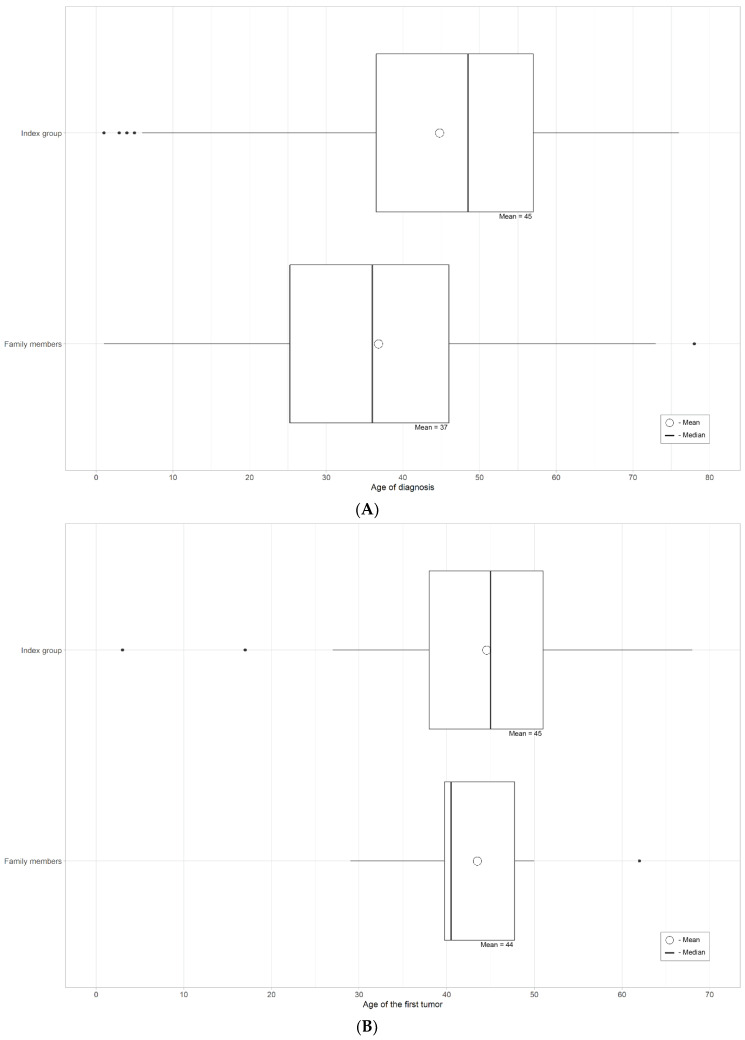
(**A**) Mean and median age of LS diagnosis in the diagnostic cohort. (**B**) Mean and median age of first tumor diagnosis in the diagnostic cohort.

**Figure 4 cancers-15-03663-f004:**
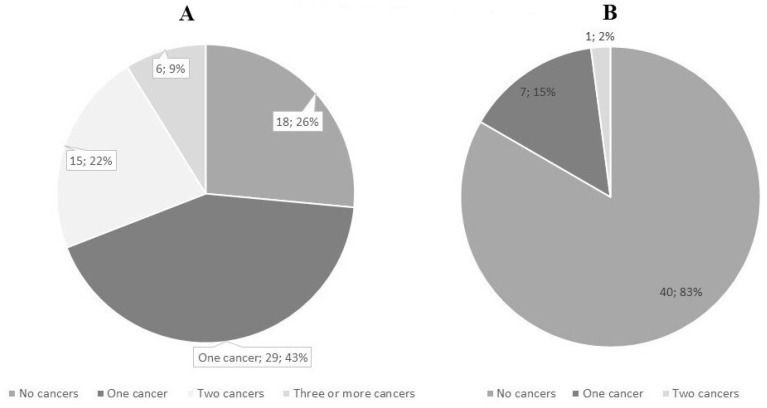
Cancer cases in health history in diagnostic cohort’s index cases (**A**) vs. family members (**B**).

**Figure 5 cancers-15-03663-f005:**
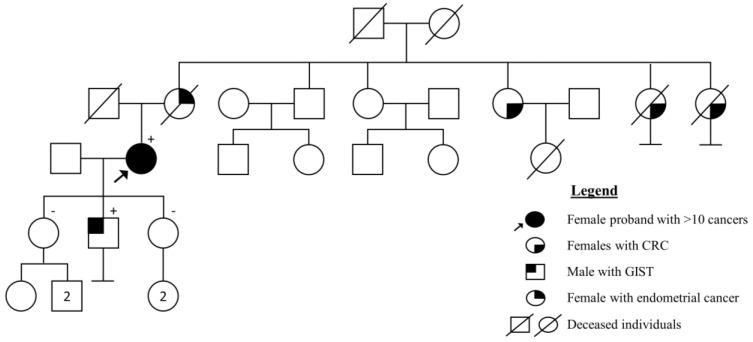
Pedigree of the case report.

**Figure 6 cancers-15-03663-f006:**
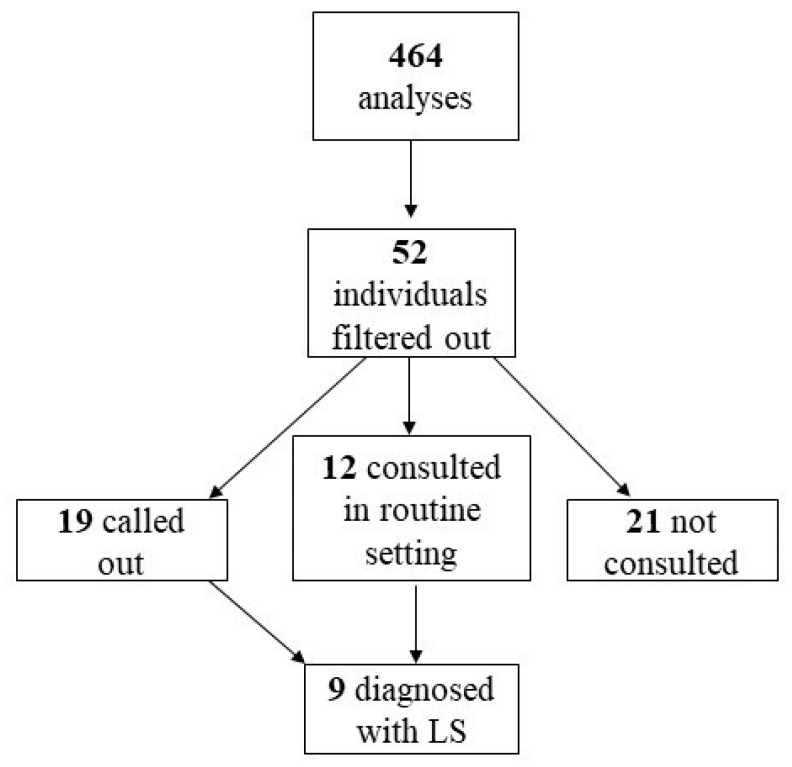
MMR IHC pilot study groups.

**Table 1 cancers-15-03663-t001:** Details of diagnostic cohorts’ cases (index cases and family members).

	Total	Not Enough Detailed Information	Healthy or Benign Changes	One Cancer in Health History	Two Cancers at Different Time or Site in Health History	Three or More Cancers in Health History	Most Frequent Cancer Site	Mean Age of First Cancer	Mean Age of LS Diagnosis	MMR IHC Performed
**Diagnostic cohort; index cases**	71	3	18	29	15	6	CRC	44.8 years	44.6 years	26/50 (52%) for cancer cases; 10/26 (38.5%) MLH1 and PMS2 negative expression
**Diagnostic cohort; family members**	48	None	40	7	1	None	CRC	43.5 years	36.8 years	2/8 (25%) for cancer cases

**Table 2 cancers-15-03663-t002:** Pathogenic and likely pathogenic variants in MMR genes detected in our LS cohort.

Gene	Variant	No. of Individuals (%)	Exon/Intron Position	Class of Variant Based on ACMG Criteria
NM_000249.4(MLH1)				
*MLH1*	c.1976G>C, p.(Arg659Pro) ‡	13 (31.7%)	17	Pathogenic
*MLH1*	c.1668-1G>T, p.? ‡	4 (9.6%)	Intron 14	Likely pathogenic
*MLH1*	c.55A>T, p.(I19F)	4 (9.6%)	1	Pathogenic
*MLH1*	c.92C>T, p.(Ala31Asp)	3 (7.32%)	1	Likely Pathogenic
*MLH1*	c.751del, p.(Tyr251Thrfs*3)	3 (7.32%)	9	Pathogenic
*MLH1*	c.1168delG, p.(Glu390Asnfs*11) †	2 (4.8%)	12	New in this study Likely pathogenic
*MLH1*	c.1918C>T, p.(Pro640Ser)	2 (4.8%)	17	Likely pathogenic
*MLH1*	c.2128_2131dupAACT, p.(Ser711*) †	1 (2.4%)	19	New in this study Likely pathogenic
*MLH1*	c.146T>A, p.(Val49Glu)	1 (2.4%)	2	Pathogenic
*MLH1*	c.840T>G, p.(Tyr280*)	1 (2.4%)	10	Pathogenic
*MLH1*	c.1685A>C p.(Gln562Pro)	1 (2.4%)	15	Likely pathogenic
**NM_000251.3(MSH2)**				
*MSH2*	c.1283_1284delAC, p.(His428Profs*14)	4 (14.8%)	8	Likely pathogenic
*MSH2*	*MSH2* exon 9–15 deletion NC_000002.11:g.(?_47690170)_(47708011_?)del †	4 (14.8%)	Deletion ex 9–15	New in this study Likely pathogenic
*MSH2*	c.793-1G>A, p.? ‡	3 (11.1%)	Intron 4	Likely pathogenic
*MSH2*	*MSH2* exon 8 deletion NC_000002.11:g.(?_47669476)_(47710098_?)del	2 (7.4%)	Deletion ex 8	Pathogenic
*MSH2*	c.1164_1165delinsGT, p.(Asn388_Arg389delinsLys*)	2 (7.4%)	7	Pathogenic
*MSH2*	c.289C>T, p.(Gln97*)	2 (7.4%)	2	Pathogenic
*MSH2*	c.1661+5G>A, p.?	1 (3.7%)	10	Likely pathogenic
*MSH2*	*MSH2* exon 11–14 deletion NC_000002.12:g.(?_47698104)_(47705658_?)del	1 (3.7%)	Deletion ex 11–14	Pathogenic
*MSH2*	c.942+3A>T, p.?	1 (3.7%)	Intron 5	Pathogenic
*MSH2*	c.942+1G>T, p.?	1 (3.7%)	Intron 5	Likely pathogenic
*MSH2*	c.181C>T, p.(Gln61*)	1 (3.7%)	1	Pathogenic
*MSH2*	c.2131C>T, p.(Arg711*) ‡	1 (3.7%)	13	Pathogenic
*MSH2*	c.1942dupA, p.(Ile648Asn*fs6) †	1 (3.7%)	12	New in this study Likely pathogenic
*MSH2+EPCAM*	*MSH2* exon 1–7 and EPCAM exon 9 deletion NC_000002.12:g.(?_47614711)_(47657080_?)del †	2 (7.4%)	*MSH2* ex. 1–7 *EPCAM* ex. 9	New in this study Likely pathogenic
*MSH2+EPCAM*	MSH2 exon 1–6 and EPCAM exon 8–9 deletion NC_000002.12:g.(?_47612305)_(47643568_?)del †	1 (3.7%)	*MSH2* ex. 1–6 *EPCAM* ex. 8–9	New in this study Likely pathogenic
**NM_000179.3(MSH6)**				
*MSH6*	c.3226C>T, p.(Arg1076Cys)	14 (43.75%)	5	Likely pathogenic
*MSH6*	c.3514dupA, p.(Arg1172Lysfs*5)	4 (12.5%)	6	Pathogenic
*MSH6*	c.2419G>T, p.(Glu807*)	3 (9.4%)	4	Pathogenic
*MSH6*	c.1998dupT, p.(Asp667*) †	3 (9.4%)	4	New in this study Likely pathogenic
*MSH6*	c.3725G>A, p.(Arg1242His)	1 (3.1%)	8	Pathogenic/Likely pathogenic
*MSH6*	c.3522dup, p.(Thr1175Tyrfs*2) †	1 (3.1%)	6	New in this study Likely pathogenic
*MSH6*	c.2308G>T, p.(Gly770Cys) †	1 (3.1%)	4	New in this study Likely pathogenic
*MSH6*	c.3261del, p.(Phe1088fs)	1 (3.1%)	5	Pathogenic
*MSH6*	c.3195_3199del, p.(Asn1065Lysfs*5)	1 (3.1%)	5	Pathogenic
*MSH6*	c.2569_2572del, p.(Asp857Phefs*10)	1 (3.1%)	4	Pathogenic
**NM_000535.7(PMS2)**				
*PMS2*	c.861_864del, p.(Arg287Serfs*19)	11 (26.2%)	8	Pathogenic
*PMS2*	c.1666del, p.(Glu556Lysfs*39) †	8 (19%)	11	New in this study Likely pathogenic
*PMS2*	c.703C>T, p.(Gln235*)	4 (9.5%)	6	Pathogenic
*PMS2*	c.2413C>T, p.(Q805*)	4 (9.5%)	14	Pathogenic
*PMS2*	c.1939A>T, p.(Lys647*)	3 (7.14%)	11	Pathogenic
*PMS2*	c.2506del, p.(Glu836Argfs*15)	2 (4.76%)	15	Pathogenic
*PMS2*	c.2445+1G>T,	2 (4.76%)	14	Pathogenic/Likely pathogenic
*PMS2*	c.2192_2196del, p.(Leu731Cysfs*3) in mosaic level 10%	2 (4.76%)	13	Pathogenic
*PMS2*	c.2T>A, (p.Met1?)	1 (2.4%)	1	Pathogenic
*PMS2*	c.634C>T, p.(Gln212*)	1 (2.4%)	6	Pathogenic
*PMS2*	c.137G>T, p.(Ser46Ile)	1 (2.4%)	2	Likely pathogenic
*PMS2*	c.319C>T, p.(Arg107Trp)	1 (2.4%)	4	Likely pathogenic
*PMS2*	c.1588C>T, p.(Gln530*)	1 (2.4%)	11	Pathogenic

† new, detected in this study. ‡ found in both diagnostic and EstBB cohorts.

## Data Availability

The data presented in this study are available on request from the corresponding author.

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
