# Peer review of "The Prevalence and Molecular Landscape of Lynch Syndrome in the Affected and General Population"

_cancers, 2023, doi:10.3390/cancers15143663_

Round 1
Reviewer 1 Report
This is original research on the prevalence of Lynch Syndrome (LS) in Estonia using data from 2 cohorts:
The cohort with detailed cancer cases obtained retrospectively and prospectively from different laboratories and the "EstBB" cohort using biobank data representing the general population of Estonia.
Different types of prevalence were calculated: Population prevalence based on diagnoses between 2012-2022 and a "birth prevalence" based on year of birth and number of live births between 1930 to 2003.
It is interesting to see the change in birth prevalence over time (1930-2020), and the increase in population prevalence over the last 10 year. I note that the birth prevalence had a peak in 1972, would a peak and decline in the population prevalence be expected in the future?
Characterisation of the diagnostic cohort is shown comparing their LS diagnosis age in the index cases with LS-diagnosis age in family members. I note that the family members diagnosis is based on the inherited genetic mutation alone, compared to the index case which diagnosis is based on both cancer and the genetic mutation.
The mutation landscape is presented with new variants and some detailed case studies. There is a different spectrum of gene mutations between the case-based cohort and the general population cohort which would be expected given the different penetrances for the MMR genes, with lower penetrance genes (MSH6 and PMS2) been detected at a higher rate in the EstBB general population data.
Finally, data from a pilot project testing tumour IHC shows an increased yield in diagnosing LS upto age 70. In this pilot study, MLH1- and PMS2- negative tumours were most common, but MSH2 gene pathogenic variants are the majority detected. Would the methylation/BRAF status for MLH1- tumours mean that many of these tumours are sporadic, but why is MSH2 variant proportion so high ( two-thirds on line 349) ?
There are some minor issues:
Line 176: "We identified 38 probands" -> It's not clear what these 38 probands are part of, there is no mention of these 38 probands again in the paper.
On line 263, it says Figure3B, but should this be Figure 4B?
Line 372-373: For PMS2, the "earlier estimations" show our population is different? What is the "recent data" to show PMS2 is similar?
Overall, this is a detailed and interesting picture of LS prevalence and mutational landscape in Estonia, and perhaps an example for other researchers to explore LS in other populations to compare and contrast.
Author Response
This is original research on the prevalence of Lynch Syndrome (LS) in Estonia using data from 2 cohorts:
The cohort with detailed cancer cases obtained retrospectively and prospectively from different laboratories and the "EstBB" cohort using biobank data representing the general population of Estonia.
Different types of prevalence were calculated: Population prevalence based on diagnoses between 2012-2022 and a "birth prevalence" based on year of birth and number of live births between 1930 to 2003.
It is interesting to see the change in birth prevalence over time (1930-2020), and the increase in population prevalence over the last 10 year. I note that the birth prevalence had a peak in 1972, would a peak and decline in the population prevalence be expected in the future? Thank you for the question, it simply means that people born in 1972 are now in their 50-s, where the probability of cancer is higher. It does not mean we expect a decline in the population prevalence in the future, we rather expect that people born after 1972 will probably develop cancer in their 40s-50s.
Characterisation of the diagnostic cohort is shown comparing their LS diagnosis age in the index cases with LS-diagnosis age in family members. I note that the family members diagnosis is based on the inherited genetic mutation alone, compared to the index case which diagnosis is based on both cancer and the genetic mutation. Thank you for the comment, the diagnosis is based on when the individual was genetically tested positive, it is the same for the index cases, but they are usually genetically tested when cancer emerges, family members are usually tested when they are (still) healthy from oncological point of view due to health awareness and sharing information in the family. According to Oxford Desk Reference (Clinical Genetics and Genomics), LS can be diagnosed solely based on the fact that an healthy individual carries a disease-causing mutation in MMR or EPCAM gene.
The mutation landscape is presented with new variants and some detailed case studies. There is a different spectrum of gene mutations between the case-based cohort and the general population cohort which would be expected given the different penetrances for the MMR genes, with lower penetrance genes (MSH6 and PMS2) been detected at a higher rate in the EstBB general population data. Thank you for the comment, I agree with you.
Finally, data from a pilot project testing tumour IHC shows an increased yield in diagnosing LS upto age 70. In this pilot study, MLH1- and PMS2- negative tumours were most common, but MSH2 gene pathogenic variants are the majority detected. Would the methylation/BRAF status for MLH1- tumours mean that many of these tumours are sporadic, but why is MSH2 variant proportion so high ( two-thirds on line 349)? Thank you for the question. Yes, indeed, most of the MLH1 and PMS2 negative tumors are sporadic. As you can see, our pilot study group was rather small, so we do not think the proportion of MSH2 is causative, or we simply do not have enough data to confirm it.
There are some minor issues:
Line 176: "We identified 38 probands" -> It's not clear what these 38 probands are part of, there is no mention of these 38 probands again in the paper. Thank you for the question. This is a mistake, and I understand it can be confusing. It should be 39 instead, I corrected it in the manuscript as well, thank you for pointing it out. This number comes from the first EstBB cohort: at first we had 30 individuals, of which we could consult 17, later it came out that there are additional 9 individuals, which makes up 39 (as one of them was deceased, and we could not recontact that individual, the final number was 38). Ee are calling out some of the additional 9 individuals together with the second EstBB cohort (those who are not already in the diagnostic cohort).
On line 263, it says Figure3B, but should this be Figure 4B?-> Thank you for the correction, it has been changed in the manuscript.
Line 372-373: For PMS2, the "earlier estimations" show our population is different? What is the "recent data" to show PMS2 is similar? Thank you for the question. It has been summed up in the manuscript: cccording to the literature, the distribution of MMR genes is as follows: MLH1 variants make up 15-40%, MSH2 20-40%, MSH6 12-35%, PMS2 5-25% and EPCAM variants <10% (Genereviews). Earlier, the contribution of MMR genes was estimated to be ~50% for MLH1, ~40% for MSH2, 7-20% for MSH6 and <5% for PMS2 [21]. Due to earlier estimations, our population is different regarding PMS2 proportion. However, recent data shows that our MMR genes distribution is similar to other populations.
Conclusively, our data showed that 23.9% of all diagnostic cohort`s index cases had disease-causing variant in PMS2, which is comparable with the data published (Genereviews, https://www.ncbi.nlm.nih.gov/books/NBK1211/). Earlier investigations showed that the proportion of PMS2 in Lynch syndrome is <5%.
Reviewer 2 Report
The manuscript is well written and is very interesting, since it represents a retro- and prospective analysis of the prevalence of Lynch syndrome in Estonian general population, describing disease-causing variants in mismatch repair genes and estimating improvement of Lynch syndrome diagnostics by raising age limit of mismatch repair genes immunohistochemistry. I think that in the Introduction more information should be added about Lynch syndrome’s molecular landscape and that in the Discussion the authors should focus more on the limitations of their study (related to the retrospective design). Moreover, there is a typo to be corrected in line 58. Concluding, I think that this manuscript is well written and needs only minor adjustments, in particular a punctuation and spelling check to eliminate typos.
The quality of English is satisfying and only minor editing of English language is required.
Author Response
Thank You for the kind review. I added some information to the introduction from molecular point of view. Also, we discussed some more of our study limitations, and this especially from the perspective of retrospective design. Also, I tried to correct all the typos and checked the punctation as well. I will upload a revised manuscript for You to take a look.